# Managing Aquifer Recharge to Overcome Overdraft in the Lower American River, California, USA

**Mahesh L. Maskey** [1,*], **Mustafa S. Dogan** [2], **Angel Santiago Fernandez-Bou** [1,3,4], **Liying Li** [1,5], **Alexander Guzman** [1], **Wyatt Arnold** [6], **Erfan Goharian** [7], **Jay R. Lund** [8] and **Josue Medellin-Azuara** [1,*]

1 Water Systems Management Lab, School of Engineering, University of California, Merced, 5200 North Lake Road, Merced, CA 95340, USA; afernandezbou@ucmerced.edu (A.S.F.-B.); lli53@ucmerced.edu (L.L.); aguzman80@ucmerced.edu (A.G.)
2 Department of Civil Engineering, College of Engineering, Aksaray University, Aksaray 68100, Turkey; msahindogan@aksaray.edu.tr
3 Sierra Nevada Research Institute, University of California, Merced, CA 95340, USA
4 SocioEnvironmental and Education Network, SEEN.team, 4 Venir Inc., Merced, CA 95340, USA
5 Environmental Systems Graduate Group, University of California, Merced, CA 95340, USA
6 Department of Civil and Environmental Engineering, University of California, Davis, One Shields Avenue, Davis, CA 95616, USA; wlarnold@ucdavis.edu
7 Civil and Environmental Engineering, College of Engineering, Columbia-Campus, University of South Carolina, Columbia, SC 2920, USA; goharian@cec.sc.edu
8 Center for Watershed Sciences, University of California, Davis, One Shields Avenue, Davis, CA 95616, USA; jrlund@ucdavis.edu
* Correspondence: mmaskey@ucmerced.edu (M.L.M.); jmedellin@ucmerced.edu (J.M.-A.)

**Abstract:** Frequent and prolonged droughts challenge groundwater sustainability in California but managing aquifer recharge can help to partially offset groundwater overdraft. Here, we use managed aquifer recharge (MAR) to examine potential benefits of adding an artificial recharge facility downstream from California's Lower American River Basin, in part to prepare for drought. We use a statewide hydroeconomic model, CALVIN, which integrates hydrology, the economics of water scarcity cost and operations, environmental flow requirements, and other operational constraints, and allocates water monthly to minimize total scarcity and operating costs. This study considers a recharge facility with unconstrained and constrained flows. The results show that adding a recharge facility increases groundwater storage, reduces groundwater overdraft, and increases hydropower without substantially impacting environmental flows. Further, artificial recharge adds economic benefits by (1) reducing the combined costs of water shortage and surface water storage and (2) by increasing hydropower revenue. This study provides a benchmark tool to evaluate the economic feasibility and water supply reliability impacts of artificial recharge in California.

**Keywords:** managed aquifer recharge; economics; CALVIN; hydroeconomic metrics; water year types; overdraft

## 1. Introduction

Groundwater is an important drinking and agricultural water supply for roughly half of the world's population, and it is paramount for adapting to more frequent and prolonged droughts exacerbated by climate change [1–5]. Increasing water use has intensified aquifer depletion worldwide, including in California's Central Valley, in the United States [6–8].

Droughts are recurrent events in California that are becoming more frequent, intense, and prolonged, as in the 2012–2016 and current droughts [9,10]. Drought impacts include decreased surface water streamflow and reserves, increased groundwater overdraft, and reduced hydropower generation. Climate models show how climate change is exacerbating drought, potentially leading to further overexploitation of aquifers in California [11]. Additionally, the decreasing rate of precipitation as snowfall and the earlier snowmelt

may increase flood runoff during wet seasons, increasing immediate surface runoff but with less infiltration to groundwater [12–16]. Reduced infiltration will hinder groundwater recharge and affect conjunctive water use operations. The 2012–2016 California drought increased the economic cost of electricity generation by a total amount of about USD 2.0 billion because of reduced hydropower generation compared to average water years (2001–2015) [17].

Northern California has more precipitation and water resources than the rest of the state, particularly during the wet season, while more water-dependent economic activities tend to occur in the central and southern regions in spring and summer [18,19]. Almost 85% of Californians rely in part on groundwater, and potential depletion of groundwater resources can be devastating for many farmers and households in the Central Valley and Central Coast [20]. Water scarcity driven by prolonged droughts may increase groundwater overdrafts in the San Joaquin Valley, ranging from 1.2 $km^3$/yr to 2.5 $km^3$/yr annually over the 2012–2016 drought years [21]. Scanlon et al. [22] estimate that about 43 $km^3$ of groundwater storage was depleted during past droughts (1976–1977; 1987–1992, 2012–2016). The 2012–2016 drought intensified groundwater pumping exacerbating overdraft of the Central Valley's groundwater basins [23,24].

To cope with climate variability, California's expansion and diversification of water storage capacity (particularly underground) and its water supply portfolio generally are promising avenues to overcome long-term groundwater overdraft [25]. The California Department of Water Resources (CA DWR) in the past few years introduced the Flood-MAR concept (managed aquifer recharge), which employs occasional excess water available during wet periods to replenish aquifers and increase groundwater reserves in preparation for droughts. Flood-MAR projects have some potential to rehabilitate and modernize California's water and flood management and infrastructure [26]. This concept can be beneficial for water supply, hydropower generation, water quality, and ecosystem enhancement [27,28].

California's Central Valley aquifer system in principle has room for more than 170 $km^3$ of recharge water (i.e., more than thrice the capacity of the state's surface water reservoirs) [29]. Out of the 160,000 $km^2$ in the Central Valley floor, at least 15,000 $km^2$ million hectares of agricultural land have favorable groundwater recharge potential [30–32]. MAR considers capturing available flood water during wet periods with low water demand and moves this water under controlled conditions into aquifers by an infiltration method. Such a framework can be helpful to meet the regulatory goals under California's Sustainable Groundwater Management Act of 2014 (SGMA), which requires groundwater basins to be in balance in extraction and recharge by 2040. Considering the long-term benefits and improved prospects for sustainability under MAR, quantifying the economic benefits of such management strategy remains largely unexplored [33]. The use of hydroeconomic models in large and complex water networks such as California's can be useful to quantify net benefits of MAR.

This study examines the economic net benefits from MAR on water storage, deliveries and hydropower production using a hydroeconomic model. Research questions include: Can a recharge facility increase hydropower generation and revenue? Can a recharge facility reduce groundwater overdraft? What are the associated economic tradeoffs from managed recharge operations?

We employ the CALVIN (California Value Integrated Network) hydroeconomic model under historical hydrology (1921–2003) to answer these questions. CALVIN is an economically driven optimization model of California that explicitly integrates the operation of water facilities, resources, and demands for California's inter-tied water supply system. Here, CALVIN is used to evaluate potential economic and water supply reliability effects from adding a recharge facility. Such effects include changes in surface and groundwater storage, hydropower generation and revenue, surface water return flows, environmental flows, water deliveries, and scarcity costs. Finally, the results from this study can provide insight for reoperating the Folsom Lake reservoir in conjunction with managed aquifer recharge for overall water supply reliability, hydropower generation, and flood control.

The rest of the paper is structured as follows. The study area and methods are described in Section 2, including a brief overview of the CALVIN model. Section 3 presents major results and relevant discussions in a broader context. Section 4 presents study limitations. Finally, Section 5 concludes with some potential future research.

## 2. Materials and Methods

### 2.1. Study Area

California's American River Basin covers 4823 km$^2$ and originates in the Sierra Nevada Mountains draining into the Sacramento River. The basin's climate has distinct wet and dry seasons, with about 90% of precipitation occurring between November and April. Annual precipitation throughout the basin ranges from roughly 500 mm to 1800 mm, with an average of about 1350 mm for the drainage area above the Folsom Dam, the major surface reservoir in the basin [34,35]. Average lake evaporation is roughly about 0.85 m$^3$/s [36]. The Folsom Reservoir is east of the Sacramento metropolitan area (Figure 1). The three forks in the upper watershed, the North, the Middle, and the South, drain to Folsom Lake. Like other river basins, both the American River Basin and Folsom Reservoir are vulnerable to climate change [37]. The federally managed Central Valley Project (CVP) oversees Folsom Reservoir operations, and in some dry years the surface water storage is released to meet downstream demands; oftentimes, at the expense of water supply reliability, dry conditions continue.

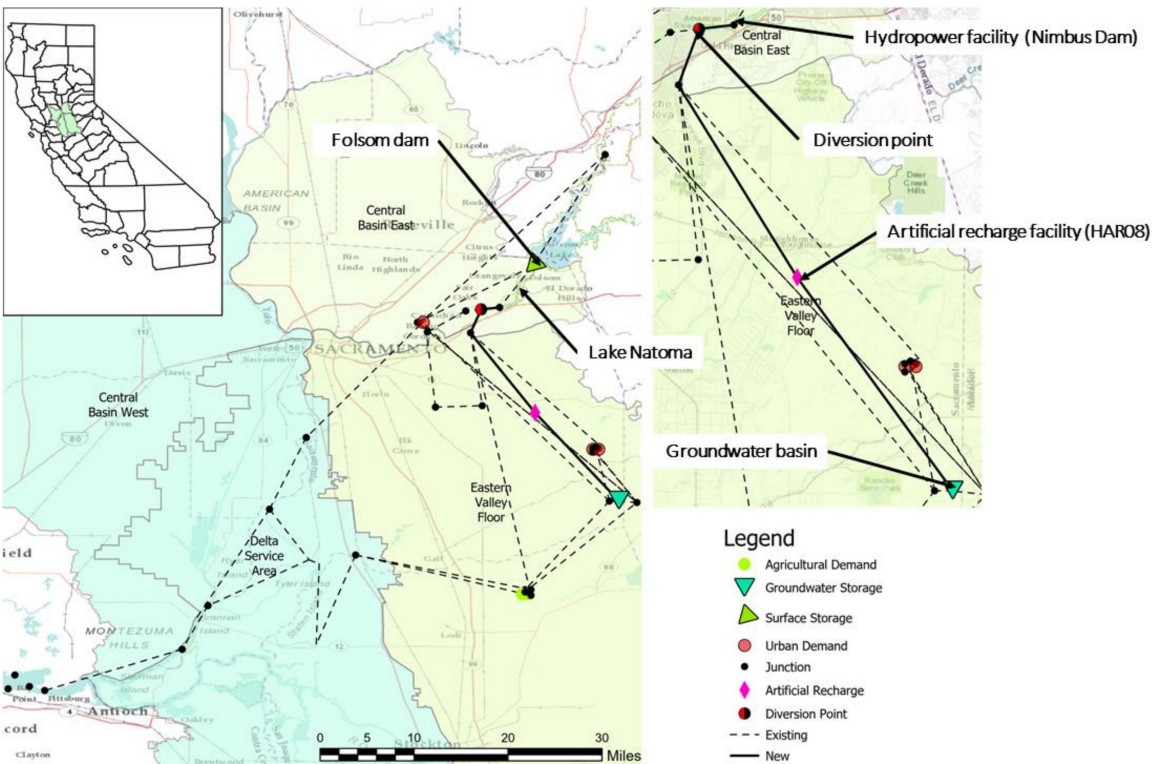

**Figure 1.** Study area map of the Folsom Lake area: this figure highlights California's inter-tied network implemented in CALVIN, including major facilities, including surface water and groundwater basins and new artificial recharge facilities connected by new conveyances, in solid black lines labeled "New."

The southern part of the Lower American River is within the CVPM (Central Valley Production Model) region 8 in CALVIN (Figure 1 and Figure S1). In the vicinity of Folsom, various water purveyors receive water from Folsom Lake, about 40.70 × 10$^{-3}$ km$^3$/year, including San Juan Water District, El Dorado Irrigation District, the City of West Sacramento, Placer County, and Roseville [38,39]. In the southern downstream service areas, existing

water infrastructure allows water deliveries to Omochumne-Hartnell Water District, Galt Irrigation District, and the City of Galt (Figure S1), including 56,600 hectares of crops [40].

The Army Corps of Engineers built Folsom Dam, a multi-purpose reservoir operated by the USBR (US Bureau of Reclamation), to minimize flood risk through the Central Valley Project. In addition to flood control and hydropower generation, the Folsom Reservoir maintains municipal and irrigation demands and supports Sacramento–San Joaquin Delta environmental flow requirements. In addition, it regulates high peak flows on the American River (more details in [18,41]). Physical water storage capacity of the Folsom Reservoir is $1.21 \times 10^{-3}$ km$^3$, but only about half of its capacity is available for flood control. The reservoir sends water to a 199 MW hydropower plant, which generates about 10% of the annual energy use in the Sacramento metropolitan area [35].

Folsom Dam operations follow a rule curve for flood control. The extreme flood events in the American River in 1986 and 1997 led to re-evaluating the river's probable maximum flood (PMF) and updating the reservoir's outlet structure and rule curve, which now allows the reservoir to empty faster at a lower elevation with a higher flood capacity downstream [35]. The downstream dam at Lake Natomas regulates the Lower American River flow for hydropower generation and diverts water towards the southern part of the watershed for water demands from the Folsom South Canal.

The Folsom South Canal transfers water to the south and the Cosumnes River Basin [29,41]. Alluvial sediments under and west of Folsom South Canal have potential for MAR, and large flows to the upstream reservoir and river system are sometimes available in wet seasons [35,40,42]. Southern groundwater storage in the American River Basin is in overdraft, impacting the endangered fall-run Chinook salmon in the Cosumnes River [43,44] and making room for augmenting the system storage in the American–Cosumnes River Basin. The Folsom Reservoir's reoperation can consider diversion of the American River flood flows towards strategic recharge locations in the American–Cosumnes River system basin and farmlands in central and southern areas [26]. In this study, we investigate the potential net benefits of adding a groundwater recharge facility via the MAR in the American–Cosumnes River system.

### 2.2. Hydroeconomic Model: CALVIN

Several hydrologic frameworks integrate surface and groundwater hydrology with economics, policy, and environmental flows to manage water resource infrastructure [45–48]. CALVIN allows the investigation of water management scenarios employing an 82 year (1921–2003) monthly historical hydrology that includes surface water, groundwater, agriculture, urban demand, and environmental requirements [49–54]. Previous work has shown CALVIN's versatility and robustness in a wide range of water management applications in California and Mexico [55]. Table S1 summarizes applications of CALVIN to date.

The CALVIN model minimizes statewide scarcity and operating costs by allocating water to agricultural and urban users monthly over the 1921–2003 period. The cost minimization objective is subject to physical, environmental, and policy constraints, including environmental flow requirements, facility capacities, and flood control operations. CALVIN integrates water resources supplies, demands, infrastructure, management policies, and economic values in a single computational platform suitable for informing a broad range of decision-makers [56]. CALVIN provides insights to suggest promising water market trades, artificial recharge (and conjunctive use operations), and alternative water use options such as desalinated, potable, and non-potable water and wastewater treatment considering both scarcity and operating costs.

A set of network nodes, N, represents storage, supply (source), and demand (sink) in the system. Links, L, connect water resource infrastructures represented by the nodes. The objective function representing the overall cost of water allocation within the entire system in monetary units is given by Equation (1) below:

$$\min Z \equiv \sum_i \sum_j \sum_k c_{ijk} X_{ijk} \tag{1}$$

$$l_{ijk} \leq X_{ijk} \leq u_{ijk}, \qquad \forall i, j, \in N, \forall k \in L \tag{2}$$

$$\sum_i \sum_k X_{jik} = \sum_i \sum_k a_{ijk} X_{ijk}, \forall j \in N \tag{3}$$

where L is a set of links defined by $(i, j, k)$. i is the origin node (supply or release node), and j is the terminal node (demand or receiving node), and k is a magnitude-related segment. $X_{ijk}$ is the decision variable in $km^3$/month for flow or storage corresponding to a time or space link and node. $c_{ijk}$ is the unit cost (USD/$km^3$) of water flowing through the link between nodes i and j within arc k, and $a_{ijk}$ is the amplitude that reflects a fraction of gain or loss from a link, including evaporation loss from the reservoir. The lower and upper bound constraints are represented by $l_{ijk}$ and $u_{ijk}$ in $km^3$/month, respectively. Equation (2) is the combination of lower and upper bound constraints, and Equation (3) is the mass balance constraint at each node and time step.

Each link $X_{ijk}$ water allocation is optimized in CALVIN either as storage in the reservoir (groundwater or surface water) or as flow in the links. Scarcity costs for agricultural and urban demands [50] are embedded in the $c_{ijk}$ terms of Equation (1). Upper bounds represent infrastructure capacity and lower bounds represent minimum environmental flow requirements in Equation (2). Such bounds are documented in [57] and appendices.

Table 1 presents basic metadata of CALVIN, including network configuration, hydrology, constraints, economic value of water, and output that are useful for decision-makers and planners.

**Table 1.** Metadata related to CALVIN configuration and fundamental input and output to the model (source: Jenkins et al. [57], https://calvin.ucdavis.edu/improving-california-water-management-optimizing-value-and-flexibility-october-2001-report, accessed on 3 March 2022).

| |
|---|
| Network configuration: |
| Node: geographical information of water-related infrastructure such as surface water and groundwater storage, point of deliveries. |
| Links: Spatial connection between two infrastructures. |
| Input hydrology: |
| Surface and subsurface inflow and reservoir storage are derived from CALSIM II and C2VSIM. |
| Constraints: |
| Facilities and capacities are based on the flood control operation. |
| Environmental flow requirements: time series of minimum and fixed flow at certain nodes representing environmental flow requirements and wildlife refuges. |
| Agricultural demand is simply the applied water at the farm level based on 2010 projected land use data. |
| Urban demand is projected water usage based on the 2050 projected urban population. |
| The economic value of water: |
| The economic value of water for agriculture comes from the Statewide Agricultural Production model, while such for urban users comes from economic demand curves for urban water use. |
| Output: |
| Shadow values represent economic costs and benefits such as willingness to pay for water use. Such benefits also include the economic value of expanding capacities. |
| Water deliveries to agricultural and urban sectors are provided by optimized inflow to the nodes that represent demand nodes. |

As listed in Table 1, CALVIN's outputs include monthly time series of shadow values (Lagrange multipliers), monthly flows, and storage for each network location. The shadow values show the marginal cost of relaxing the constraints in each link or storage node. These values reflect the approximate economic value of incremental loosening or tightening of a constraint at each time step. Shadow values of lower and upper bound constraints inform the cost or benefit of adding one unit to water delivery conveyance or reservoir storage capacity, respectively. Negative marginal values recommend additional storage to improve the system's ability to manage water supply; "zero" implies nonbinding constraints. Finally,

CALVIN's objective function quantifies water scarcity and cost as the imbalance between supply and demand and the economic cost of water scarcity [50,57].

The CALVIN model recently was migrated to Python and started using Pyomo (Python Optimization Modeling Objects) solvers [52,58,59]. This study employs the GLPK (GNU Linear Programming Kit) solver to solve the entire water network implemented in the CALVIN with limited foresight runs. Limited foresight corresponds to a series of sequential twelve-month runs throughout analysis instead of a single run for the period of analysis. It uses carryover value functions to ensure adequate carryover storage [60,61].

### 2.3. Modeling Approach

### 2.3.1. Network Modification

Figure 2 shows a simplified network diagram of the American River Basin (Table S2). This study modifies the existing California Water Network implemented in CALVIN [62], adding a hypothetical recharge facility (pink diamond in Figure 1), HAR08, a square rhombus in Figure 2. While the diversion node, D85, below the Nimbus Dam, diverts the water to northern water users near Placer County, Folsom South Canal (D9-C173) sends water to the proposed recharge facility, HAR08, near Lodi, Bachelor Valley, and Elk Grove. Originally, junction D9 represents the hydropower plant at Lake Natomas and directly connects the South Folsom Canal. The modification includes a diversion node (DP9) before Folsom South Canal (C173) to avoid impacting the hydropower generation from Nimbus Dam (D9). DP9 diverts tailwater from the hydropower plant at D9. The recharge facility, HAR08, directly links to the groundwater basin corresponding to CVPM region 8, GW_08, shown in Figure 2.

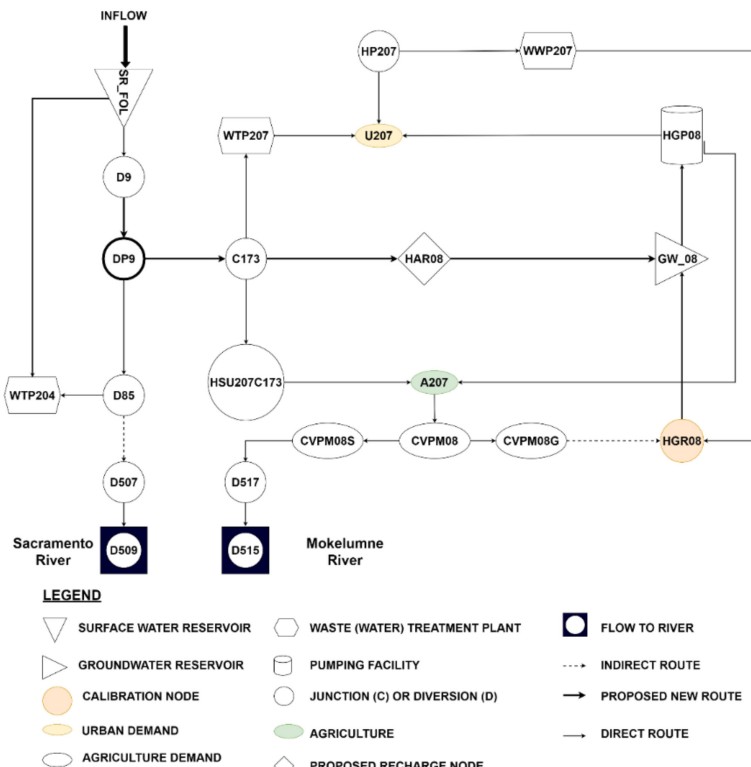

**Figure 2.** The California Water Network portion in the Lower American River Basin, south of Folsom Lake.

Based on existing losses in the CALVIN model [63,64], link amplitudes (used to represent either flow gains or losses such as recycling or canal seepage) of the upstream conveyance link C173-HAR08 and the downstream conveyance link HAR08-GW_08 are set to 0.95, allowing 5% conveyance losses to evaporation and canal seepage. Water delivery to

the recharge and groundwater facilities are assumed to occur at no cost, including capacity (physical) constraints (Table 2).

**Table 2.** Scenarios and network features for the American River water supply network. NOB refers to no bound, and $Q_m$ in km$^3$/month for month $m$.

| Scenarios * | Upstream | | Downstream | |
|---|---|---|---|---|
| | **Upper Bound** | **Lower Bound** | **Upper Bound** | **Lower Bound** |
| Base case | | | | |
| Recharge I | NOB | NOB | NOB | NOB |
| Recharge II | $Q_m$ | $Q_m$ | NOB | NOB |
| Recharge III | $Q_m$ | NOB | NOB | NOB |
| Recharge IV | $Q_m$ | NOB | $Q_m$ | NOB |

Notes: * Base case has no recharge facility. Recharge I (first scenario) refers to unconstrained upstream and downstream conveyance to the recharge facility. Recharge II (second scenario) considers only unconstrained downstream conveyance with equal bound at the upstream conveyance facility. Recharge III (third scenario) has constrained upper bound upstream and unconstrained lower bound upstream and downstream conveyance. Recharge IV (fourth scenario) has constrained upper bound upstream and downstream and unconstrained lower bound upstream and downstream.

Monthly inflow to the recharge facility ($Q_m$ for the recharge months, November to March) is reported in Table 3. About 90% of precipitation falls above Folsom Lake between November and April. Consequently, more water (averaging about 225 km$^3$/yr) is available for winter (November–March) recharge periods: NOB (no bound) = 1000 km$^3$/month for recharge months in normal and wet years; NOB = 0 for all other years.

**Table 3.** Statistics of monthly averaged optimized recharge flow. Flows in km$^3$/month from November to March over 1993–2003 are derived from the surrogate model [35].

| Month | November | December | January | February | March |
|---|---|---|---|---|---|
| Minimum, $Q_{mn}$ | 3.53 | 3.18 | 5.09 | 5.88 | 36.50 |
| Average, $Q_m$ | 41.77 | 43.58 | 44.95 | 44.20 | 50.28 |
| Maximum, $Q_{mx}$ | 54.42 | 54.42 | 54.42 | 54.42 | 54.42 |

Table 2 lists the combination of constrained and unconstrained bounds upstream and downstream for each analyzed scenario (except for the base case historical run). In the constrained case, either the upper or lower bounds are optimized recharge values from a surrogate model, and CALVIN is forced to use that flow (Table 2). In the unconstrained cases, these bounds are set free to optimize flow in the recharge link.

CALVIN allows using a carryover storage value, connecting multiple limited foresight runs in a sequence [60,61]. Lack of such values produces less realistic water allocation decisions under limited foresight, particularly running storage as low as possible each year. Arnold [61] suggests penalizing ending storage of surface water reservoirs and groundwater storage. The ending groundwater storage is penalized at USD 0.08/m$^3$ while ending surface water storage is penalized at USD 0.73/m$^3$ [61]. The storage penalties are derived from the respective shadow values on storage of surrounding facilities implied by the base case run. The average monthly shadow value is considered nearest node D9 for surface water and HGP08 for groundwater storage.

To reflect the changes in surface water storage at the end of the period, we keep initial storage the same as initial default storage for the first year. For subsequent years, the ending storage is transferred from the previous year ($t − 1$) as initial storage:

$$S_{initial}(t) = \begin{cases} S_{SW} & if\ t = 1 \\ S_{end}(t-1) & if\ t > 1 \in N \end{cases} \tag{4}$$

where $S_{SW}$ is the initial default storage (km$^3$), $S_{end}(t-1)$ is the carryover ending storage for the year $t-1$ (km$^3$), and $N$ is the number of simulation years. In each run, ending storage is constrained to long-term fixed storage value so that the change in storage at the end of each year can be assessed without ending at $S_{SW}$.

Likewise, constraining the groundwater storage at the end of the period sets the initial storage as maximum groundwater storage, $S_{GW}$ (km$^3$), for the first period and transferred ending storage (km$^3$) from the previous year $(t-1)$ as:

$$S_{initial}(t) = \begin{cases} S_{GW} & if \ t = 1 \\ S_{end}(t-1) & if \ t > 1 \in N \end{cases} \tag{5}$$

In this case, storage at the end of the year is forced to change by setting ending storage as equal to the assumed maximum groundwater storage as $S_{end}(t) = S_{GW}$. It helps us visualize how groundwater storage changes, reflecting elimination of overdraft.

### 2.3.2. System Analysis

The hydropower plant at Nimbus Dam—diversion D9 (Figure 3)—employs a piecewise linear function given GLPK is a linear solver that cannot solve nonlinear functions directly, such as those involving flow storage and head in hydropower generation [51,65,66]. Then, hydropower generation estimates from the Folsom Dam involves post-processing the optimized storage and release for release, storage, and hydropower price in CALVIN using Equation (6) below:

$$P_t = \rho \ g \ \eta \ H(S_t) \ Q_t \tag{6}$$

where $P_t$ (W) is total hydropower output at time t; $\rho$ (kg/m$^3$) is the density of water; $g$ (m/s$^2$) is gravitational constant; $\eta$ is the plant efficiency; $H(s_t)$ in m is the water head for the month $t$; $Q_t$ is the release from the reservoir to the turbine (m$^3$/s). Dogan [66] establishes some empirical relationships between storage ($S$) and head for 11 hydropower plants of the Central Valley, by considering the nonlinearities in reservoir storage versus power generation capacity. Among them, this study uses the empirical Equation (7) for the Folsom Dam for the head dependent on storage:

$$H(S_t) = \alpha \ S_t^3 + \beta \ S_t^2 + \gamma \ S_t + c \tag{7}$$

where $\alpha = 3.21 \times 10^{-8}$, $\beta = -8.67 \times 10^{-5}$, $\gamma = 0.10$, and $c = 51$ are dimensionless regression coefficients.

The hourly energy prices in USD/MWh are from CAISO [67] for 2010 to 2018 to derive hourly hydropower revenue time series [65].

The monthly time series of storage, water delivery, water shortage, and scarcity costs in the Folsom Dam service areas are post-processed and summarized. This study does not consider climate change impacts on the system. However, the historical (1921–2003) time series includes various multi-year droughts and wet years [68].

The calculation of economic value for surface water storage assumes the cost of losing water from the reservoir [69] through evaporation. Among various losses, the evaporative loss is considered a proxy for the opportunity cost of losing water. As suggested by Arshad et al. [69], we employ the average of the upper duals or Lagrange multipliers at the upper bound ($\bar{\lambda}$ in USD/m$^3$) of downstream nodes D9 and WTP204 from the Folsom Reservoir (SR_FOL). For simplicity, the cost of surface water storage in km$^3$ is the product of the average monthly upper dual or shadow value ($\bar{\lambda}$) and evaporation ($E$ in km$^3$) from the reservoir:

$$C_{SW} = \bar{\lambda} \times E \tag{8}$$

Note that the average values of upper duals in the downstream nodes are considered for the average of either dry, normal, or wet years.

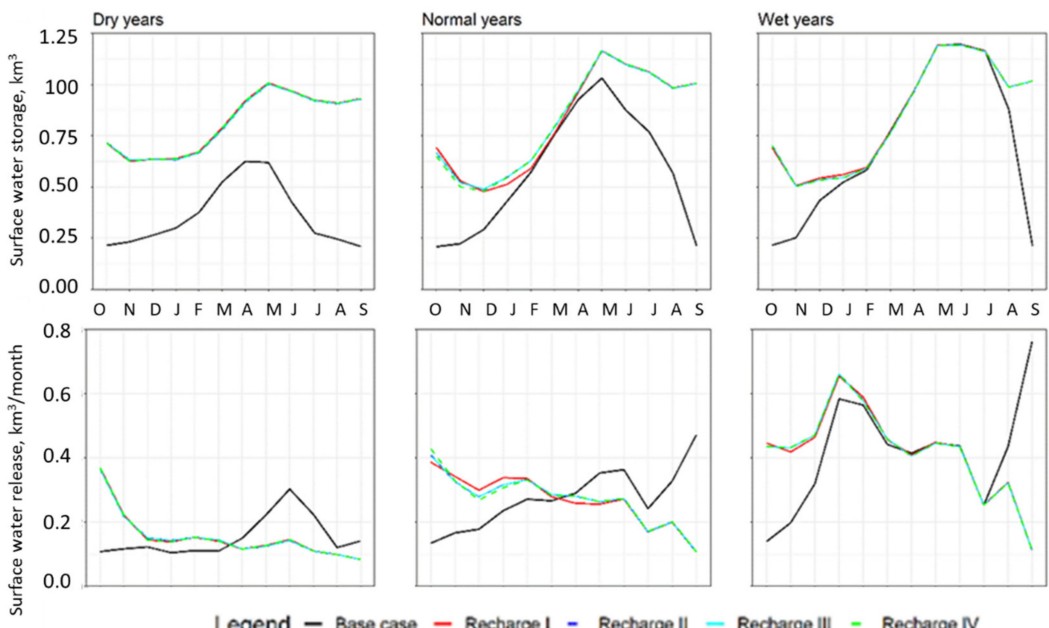

**Figure 3.** The monthly average storage behind the Folsom Dam (**top**) and average monthly release from the dam towards the hydropower facility (**bottom**) under three recharge scenarios in dry, normal, and wet years.

We explore the water delivery and scarcity costs of agriculture water supply, hydropower revenue, and surface water storage cost. In this exercise, hydropower revenue is derived once hydropower generation is calculated using Equation (6) multiplied by the monthly energy price described above. Surface water storage cost is derived from Equation (8). The agricultural water scarcity cost is the product of the unit cost of delivery links and the difference between flow in the link and target delivery derived from the Statewide Agriculture Production model [70,71]. We multiplied the unit cost with the flow value in the link for all other delivery and operation costs.

## 3. Results and Discussion

### 3.1. Changes in Surface Water Reservoir Operation

The modified network with an additional recharge facility shows an increase in surface water storage in the Folsom Dam (Figure 3, top) compared to the base condition. The greatest increased storage occurs in dry years, followed by normal years. Normal water years show increased storage from late spring to winter, while wet years show increased storage in late summer and fall. The relative surface water storage is high in dry years with the recharge facility. The increase in storage in wet years is less than in dry and normal years. This result considers costs of groundwater pumping water versus using surface storage releases. Adding an MAR facility increases surface water storage in Folsom Lake by 0.81 km$^3$/year, 0.83 km$^3$/year, and 0.85 km$^3$/year during dry, normal, and wet years, respectively.

Compared to the base case without a recharge facility, all managed recharge scenarios show an increase in the release from the reservoir from October to March, and then the release becomes decreased for the rest of the year with respect to the base case (Figure 3, bottom). Hence, Folsom Lake will be releasing more water during recharge months. Results show an increased release of 15.60 hm$^3$/yr during dry and 14.40 hm$^3$/yr during wet years; during normal years, such release is reduced by 55.00 hm$^3$/yr with respect to the baseline condition.

A marginal value in surface water storage refers to the shadow value at the lake from mass balance constraints (Equation (3)), while opportunity cost refers to the shadow value associated with upper bound constraints (Equation (2)). Table S3 in the Supplemental

Material shows both values. The values imply decreased marginal values and opportunity costs from surface water storage in dry and normal years but increased in wet years. In dry years, marginal values decreased from USD 2.33/m$^3$ to USD 0.50/m$^3$, and opportunity cost from USD 0.20/m$^3$ to USD 0.08/m$^3$, suggesting a decrease in the need to expand surface water storage in Folsom. In wet years, both marginal values and opportunity cost are slightly increased to USD 0.06/m$^3$ for the ending storage. Increased opportunity costs (upper bound shadow values) in wet years hint to increased reservoir capacity to store more water during wet years.

### 3.2. Hydropower Benefit

Increased surface water storage and release during recharge months (October to March) also increases hydropower generation and revenue compared to baseline conditions (Figure 4). Nevertheless, there are decreases in hydropower generation for the non-recharge months (Figure 4, top). Dry years also show increased hydropower generation in August. Hydropower revenues closely follows trends of hydropower generation (Figure 4, bottom). Such an increase in hydropower generation is shown in Table 4, with a recharge facility increased hydropower estimated as 85 MWh/yr in dry years, 53 MWh/yr in normal years, and 66 MWh/yr in wet years. Annual hydropower revenue is also increased as much as two million USD (MUSD) per year in dry years compared to dry years under the base case. Both Recharge II and Recharge III alternatives have similar hydropower generation trends and show the highest benefit from a hydropower facility compared to the base case.

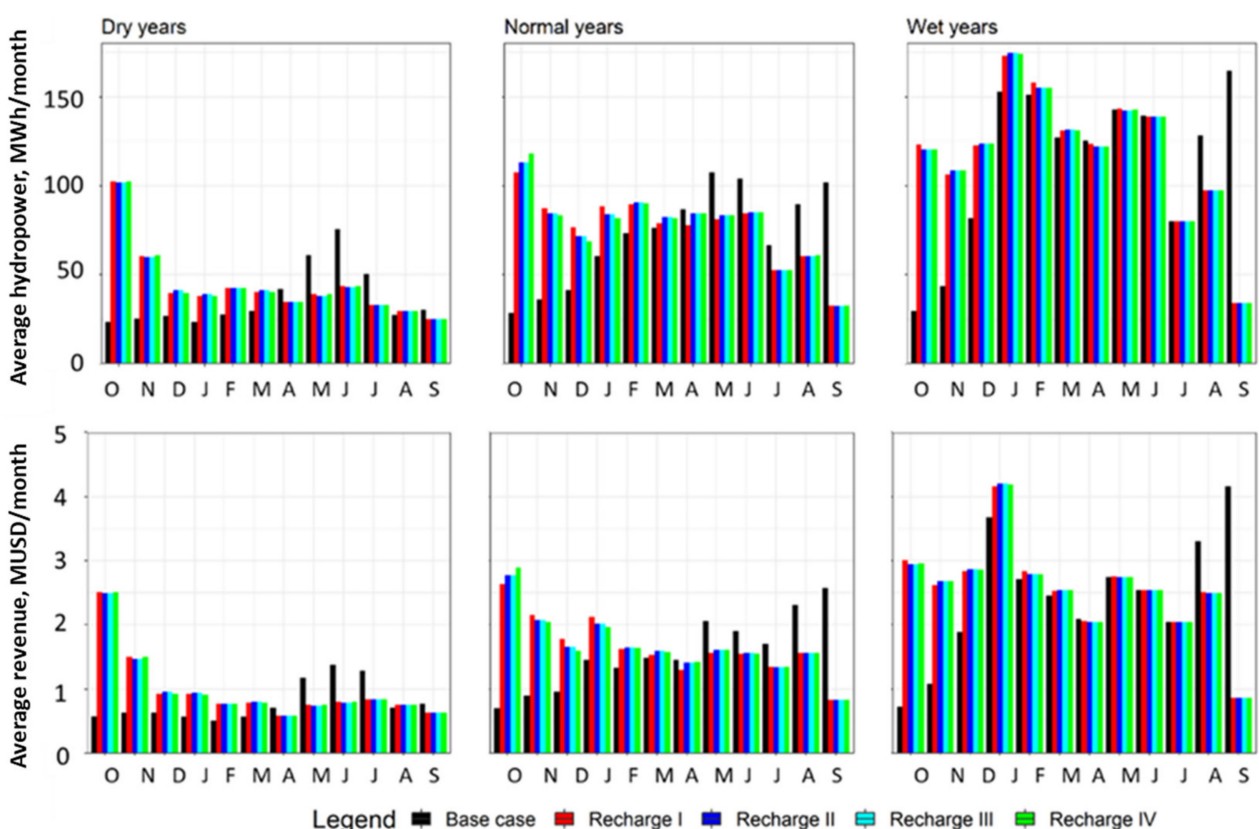

**Figure 4.** The monthly average hydropower generation (**top**) and revenue (**bottom**) from the Folsom Dam under three recharge scenarios in dry, normal, and wet years.

**Table 4.** Annual average hydropower generation and revenue (million US dollars, MUSD) over different water years.

| Scenarios | Dry Years | | Normal Years | | Wet Years | |
|---|---|---|---|---|---|---|
| | Hydropower, MWh/yr | Revenue, MUSD/yr | Hydropower, MWh/yr | Revenue, MUSD/yr | Hydropower, MWh/yr | Revenue, MUSD/yr |
| Base case | 444.20 | 10.13 | 872.38 | 20.66 | 1365.87 | 33.43 |
| Recharge I | 529.72 | 12.68 | 918.35 | 22.28 | 1431.62 | 35.12 |
| Recharge II | 529.83 | 12.71 | 925.81 | 22.41 | 1429.34 | 35.08 |
| Recharge III | 529.83 | 12.71 | 925.81 | 22.41 | 1429.34 | 35.08 |
| Recharge IV | 529.63 | 12.68 | 923.61 | 22.33 | 1428.50 | 35.05 |

### 3.3. Artificial Recharge Facility

Figure 5 depicts total recharge flow to groundwater in the study area by recharge scenario. Total recharge flow is the sum of deep percolation at the groundwater recharge facility and agricultural return flow diverted from the Folsom South Canal to the proposed artificial recharge. The wetter the climate is the more water is conveyed to the recharge facility. Bottom bar charts show estimated deep percolation at the groundwater facility, while the hatched top bars imply the additional flow due to the recharge facility. Flow in the upstream link of the recharge facility is the same as the flow represented by hatched bars due to mass balance.

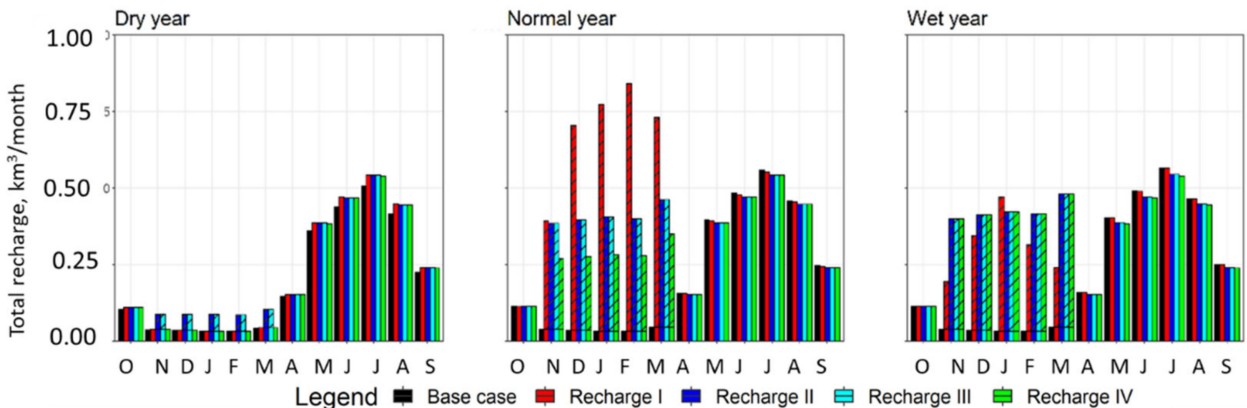

**Figure 5.** Monthly total recharge to the groundwater basin in different dry, normal, and wet scenarios. Hatched portion represents the additional recharge flow from the recharge facility, HAR08.

The unconstrained scenario (Recharge I) does not show any flow in the upstream link in dry years, as CALVIN does not identify those as cost minimizing for the scenario given more costly demands elsewhere. All flow from the surrogate model (Table 3) appears in the upstream link in wet years. Scenarios with no bound at the downstream conveyance facilities are similar, with little increased recharge flow in all years. Recharge scenario (unconstrained upstream and downstream) seems more efficient as recharge flow is more than the surrogate model optimized flow for normal and wet years, although there is no recharge flow in dry years. Consequently, the groundwater basin receives higher deep percolation flow in the dry years than in normal and wet years (Table S4), while the system obtains additional water from the recharge facility as high as 0.03 km$^3$/year in normal years and 0.02 km$^3$/year in wet years.

Table 5 shows higher annual shadow values of upstream and downstream water conveyance infrastructure to the recharge facility from the diversion node. This suggests that there are times when capacity expansions of conveyance facilities might be economically worthwhile. Further, the shadow values of the upstream conveyance facility to the recharge facility are always higher than those of the downstream facility, suggesting that it is more beneficial to expand water conveyance capacity upstream than downstream for scenarios I,

II, and III. The upper dual values at the recharge link are higher in the dry years (about USD 0.99/m$^3$) than in wet years, about USD 0.70/m$^3$ (Table 6), suggesting expanding the recharge conveyance is more beneficial in dry years.

**Table 5.** The annual average economic value of conveyance infrastructure expansions upstream and downstream as shadow values of upper bound constraints on conveyance links.

| Scenarios | Upstream, USD/m$^3$ | | | Downstream, USD/m$^3$ | | |
|---|---|---|---|---|---|---|
| | Dry | Normal | Wet | Dry | Normal | Wet |
| Recharge I | 0.99 | 0.72 | 0.69 | 0.02 | 0.01 | <0.01 |
| Recharge II | 0.99 | 0.75 | 0.70 | 0.00 | 0.00 | 0.00 |
| Recharge III | 0.99 | 0.75 | 0.70 | 0.00 | 0.00 | 0.00 |
| Recharge IV | 0.02 | 0.01 | 0.00 | 0.99 | 0.75 | 0.72 |

**Table 6.** Long-term change in groundwater storage over 82 years for different scenarios.

| | Base Case | Recharge | | | |
|---|---|---|---|---|---|
| | | I | II | III | IV |
| Ending storage, km$^3$/yr | 225.52 | 236.49 | 236.87 | 236.87 | 235.39 |
| Long-term change, km$^3$ | N/A | 10.98 | 11.35 | 11.35 | 9.88 |
| Annual change, km$^3$ | N/A | 0.13 | 0.14 | 0.14 | 0.12 |

### 3.4. Groundwater Operation

The time series of monthly groundwater storage (Figure 6, top) followed by groundwater pumping (Figure 6, bottom) over 82 years for four scenarios shows that the Recharge I scenario (unconstrained) is the most beneficial, as it recharges the aquifer after the second drought and then stabilizes groundwater storage. All other recharge scenarios show a monotonic increase in groundwater storage until the third to last drought. At the end of every dry year, there is a decrease in storage, leading to increased pumping. The unbounded recharge (Recharge I) remains relatively stable until 1994 and then starts to increase monotonically. Although the fourth case (Recharge IV, without lower bound) shows a monotonic increase after 1950, its final storage does not exceed the initial storage. As a result, groundwater storage in dry, normal, and wet years increased by 8.14 km$^3$, 7.77 km$^3$, and 9.37 km$^3$, respectively, in the Recharge I scenario. Recharge IV shows average yearly increases in groundwater storage by 5.43 km$^3$, 4.19 km$^3$, and 5.80 km$^3$ in dry, normal, and wet years, respectively. Overall, the considered groundwater basin receives increased storage by as high as 11.35 km$^3$ in the long-term scenario and as high as 0.14 km$^3$ annually (Table 6).

Increases in groundwater storage make more water available for pumping (Figure 6, bottom). The Supplemental Material in Figure S2 shows more differences in groundwater pumping between normal and wet years including decreased pumping with respect to the base case. It suggests the proposed recharge facility improves the system by making more water available during dry years at the expense of decreased pumping during normal and wet years to bank enough water underground.

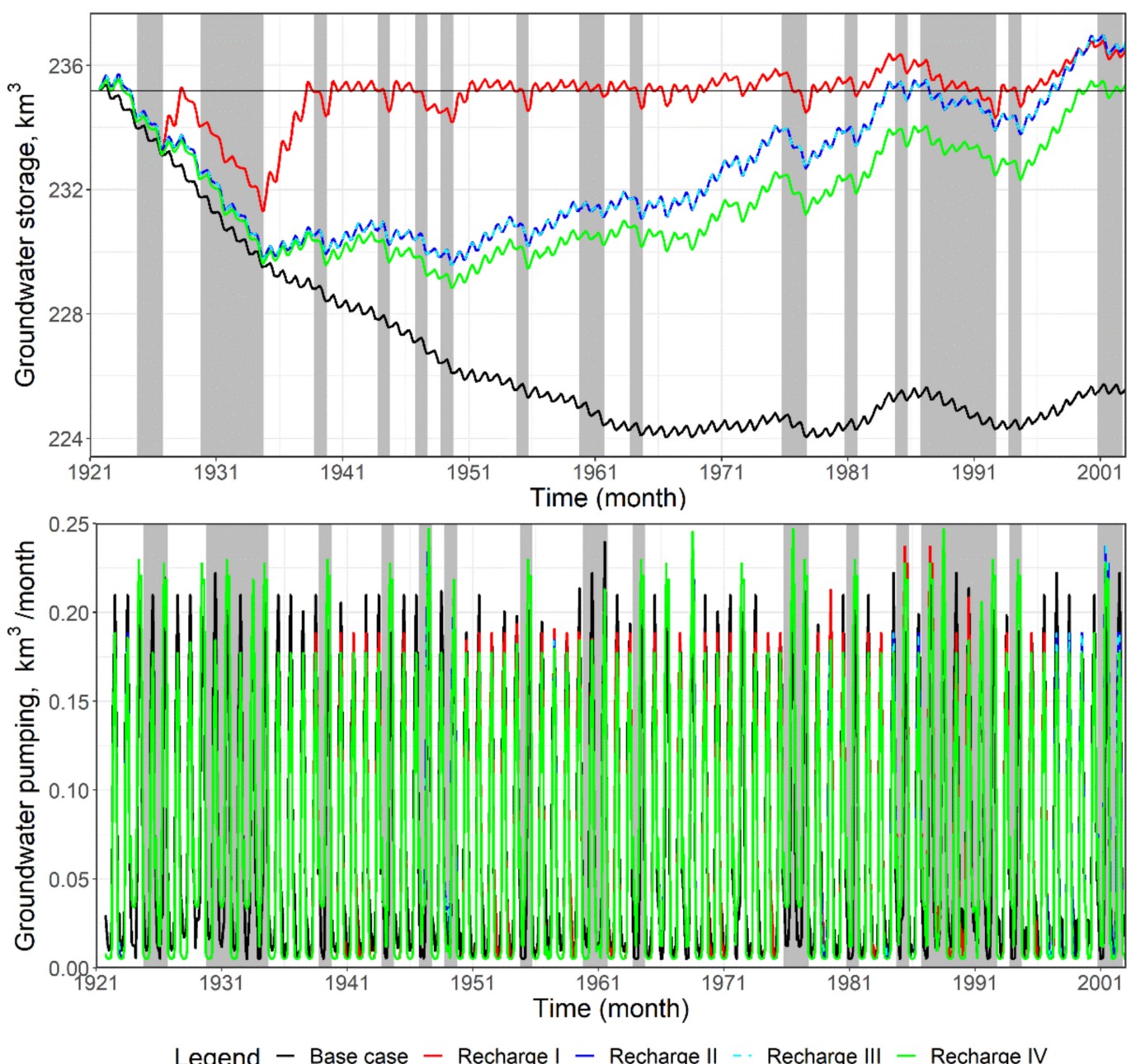

**Figure 6.** Time series of groundwater storage dynamics (**top**) and groundwater pumping (**bottom**) in three recharge scenarios. Dry years are shaded areas.

### 3.5. Agricultural Water Deliveries

Among several benefits from managed aquifer recharge, an additional recharge facility provides an avenue for improving prospects for groundwater sustainability and water transmission facilities throughout an aquifer to users' wells. Table 7 reports monthly total water deliveries to the agricultural region in the southern part of Folsom Dam. Some shortages (failure to meet target demands) are seen in the modeling results. The recharge facility increased the delivery in summer and early fall but decreased during recharge months and winter when there is no demand for irrigation water.

**Table 7.** A monthly average delivery to the agricultural region near the groundwater basin at CVPM8 and its target delivery in $hm^3$.

| Month | Base Case | Recharge I | Recharge II | Recharge III | Recharge IV | Target |
|---|---|---|---|---|---|---|
| October | 5.3 | 9.7 | 9.6 | 9.6 | 9.6 | 34.2 |
| November | 0.2 | 0.6 | 0.6 | 0.6 | 0.6 | 2.2 |
| December | 4.7 | 4.2 | 4.2 | 4.2 | 4.2 | 0.7 |
| January | 48.7 | 45.6 | 45.3 | 45.3 | 45.5 | 0.2 |
| February | 43.2 | 35.5 | 35.0 | 35.0 | 35.5 | 0.2 |
| March | 42.8 | 35.2 | 35.0 | 35.0 | 35.3 | 5.1 |
| April | 41.8 | 35.4 | 35.5 | 35.5 | 35.6 | 53.4 |
| May | 47.1 | 50.2 | 49.2 | 49.2 | 49.0 | 156.7 |
| June | 144.4 | 145.8 | 143.1 | 143.1 | 142.1 | 193.5 |
| July | 167.9 | 176.1 | 172.4 | 172.4 | 171.5 | 225.6 |
| August | 173.9 | 185.6 | 181.2 | 181.2 | 180.1 | 182.9 |
| September | 133.7 | 143.3 | 139.9 | 139.9 | 138.9 | 91.8 |
| Total | 853.8 | 867.4 | 851.1 | 851.1 | 847.9 | 946.4 |

Scenarios with the recharge facility increase the groundwater delivery to the agriculture region and decrease the surface water diversions. In summary, agriculture delivery is increased in dry but decreased in normal and wet years in all scenarios (see Figure S3).

### 3.6. Surface Water Return Flows Draining to the Delta

The impact of adding recharge facilities on Delta outflow near the Sacramento River, where it meets with the San Joaquin River, is also assessed. We investigate the monthly return flows to the Sacramento River through the link, D517–D515 (Figure 7, above), and Mokelumne River below the American River via D507–D509 river reach (Figure 7, bottom). As shown in Figure 7, the Mokelumne River receives the same flow as the base case in most months. However, the Sacramento River receives slightly less water. In addition, a flow decrease is noticed from July to September in normal and wet years. There is an increase in return flow towards the Mokelumne River by 12% and 1.6% in dry and normal years, respectively, but there is a slight decrease in wet years by 0.2%. However, the Sacramento River receives less water throughout the simulation period by 1.8%, 1.9%, and 2.5% in dry, normal, and wet years, respectively (Table 8). Overall, the addition of a recharge facility has a relatively small effect on the average return flows over the modeled period (Figure 7) and has minor negative effects considering each year's conditions to both rivers (from increased 12% return flows in dry years in the Mokelumne River to decreased 2.5% return flows in wet years in the Sacramento River). This also implies impacts on the Delta outflow and ecosystems could be minor (Figure 7, Table 8).

**Table 8.** Annual average change, $hm^3/yr$ (change in % with respect to base case), in surface water return flow towards Delta in the confluence of Sacramento and Mokelumne River for different water year types.

| Scenario | Sacramento River | | | Mokelumne River | | |
|---|---|---|---|---|---|---|
| | Dry | Normal | Wet | Dry | Normal | Wet |
| Base case | 6360 | 14780 | 35520 | 240 | 880 | 1740 |
| Recharge I | −101 (−1.6%) | −285 (−1.9%) | −560 (−1.7%) | 28 (+11.7%) | 16 (+1.8%) | 0 (0.0%) |
| Recharge II | −132 (−2.1%) | −89 (−0.6%) | −741 (−2.3%) | 28 (+12.0%) | 14 (+1.6%) | −3 (−0.2%) |
| Recharge III | −132 (−2.1%) | −89 (−0.6%) | −741 (−2.3%) | 28 (+12.0%) | 14 (+1.6%) | −3 (−0.2%) |
| Recharge IV | −113 (−1.8%) | −91 (−0.6%) | −807 (−2.5%) | 28 (+11.5%) | 14 (+1.6%) | −4 (−0.2%) |

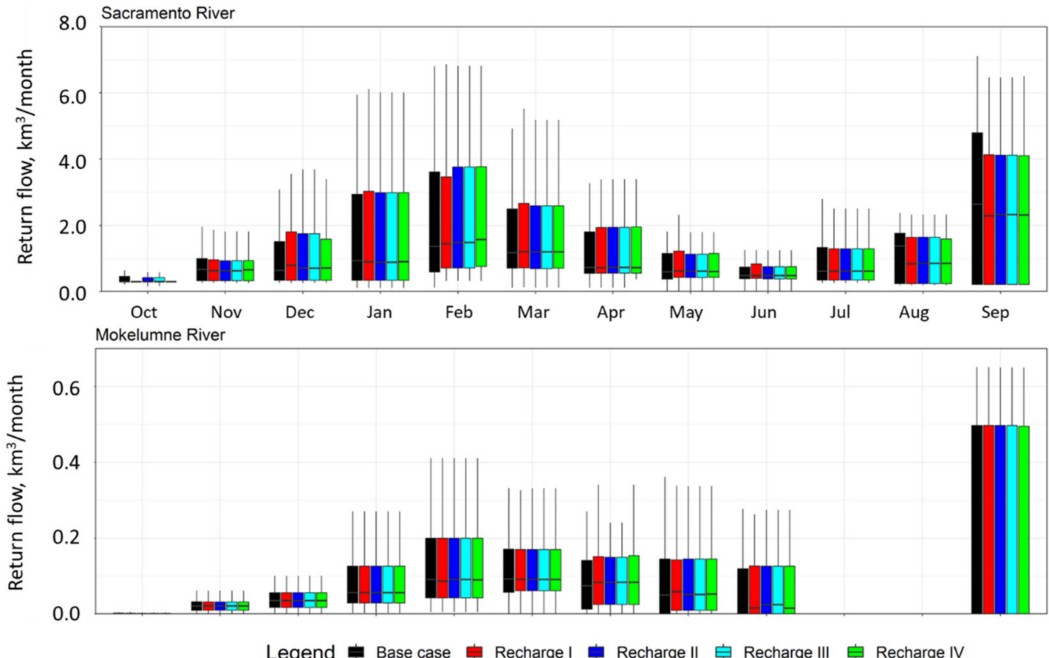

**Figure 7.** Monthly surface water return flows to Sacramento River (**top**) and Mokelumne River (**bottom**), showing the impact of recharge flow.

Further, CALVIN provides information about the economics of water allocation to maintain environmentally safe return flows at various locations downstream from reservoirs. Table 9 reports the annual average economic cost, i.e., upper bound shadow values (USD/m$^3$) of return flows at Mokelumne River near Rio Vista and Sacramento River near the Delta over different water years in 82 years of simulation. Recharge scenarios show a minimal increase in the opportunity costs of maintaining instream flow at Rio Vista, which transports a large flow Volume into the Bay of San Francisco north to south [72].

**Table 9.** Annual average upper bound shadow values (USD/m$^3$) at Mokelumne and Sacramento Rivers where minimum instream flow needs to be maintained.

| Scenario | Sacramento River | | | Mokelumne River | | |
|---|---|---|---|---|---|---|
| | Dry | Normal | Wet | Dry | Normal | Wet |
| Base case | 0.22 | 0.03 | <0.01 | 0.17 | 0.04 | 0.01 |
| Recharge I | 0.19 | 0.03 | <0.01 | 0.14 | 0.04 | 0.01 |
| Recharge II | 0.20 | 0.03 | <0.01 | 0.14 | 0.04 | 0.01 |
| Recharge III | 0.20 | 0.03 | <0.01 | 0.14 | 0.04 | 0.01 |
| Recharge IV | 0.19 | 0.02 | <0.01 | 0.14 | 0.04 | 0.01 |

*3.7. Mass Balance in the Network*

Our model runs consist of the entire California Network implemented in CALVIN, yet the only section that is analyzed is the American River which was selected for this research (Figure 2). As an illustration, Tables S5 and S6 report details of water balance at the Folsom Dam (SR_FOL) and groundwater basin (GW_08) for the drought year 1987–1988, with a focus on base case and recharge case (Recharge IV), respectively.

Outflow and storage in the recharging case are higher than in the base scenario (see Supplemental Material Table S5). This suggests that the imposed penalty at the upper bound of a storage facility that precludes changes in storage is mostly less in the recharge facility (Equation (4)). It is also noted that the higher the storage, the bigger the evaporation rate. Like increased surface water storage in the Folsom Dam, groundwater storage and

outflow, i.e., pumping, are also always high in the recharging case (Table S6). It explains the imposed penalty at the end of the year (Equation (5)).

### 3.8. Water Delivery Cost Analysis

This study also explores the regional water supply and water allocation cost portfolios to determine the potential benefits of artificial recharge below the American River. Figure 8 (top) shows the changes in four water indicators within the vicinity of Folsom, namely: hydropower flow, agricultural and urban delivery, and scarcity volume. As seen, hydropower generation is not impacted by the recharge facility in dry and wet years, but there is a slight decrease in normal years. Total agricultural water delivery from surface water and groundwater sources is increased in the recharge scenarios only during dry years, while urban users receive the same amount of water with or without the recharge facility. Importantly, water scarcity Volume decreases in dry years but not in normal and wet years.

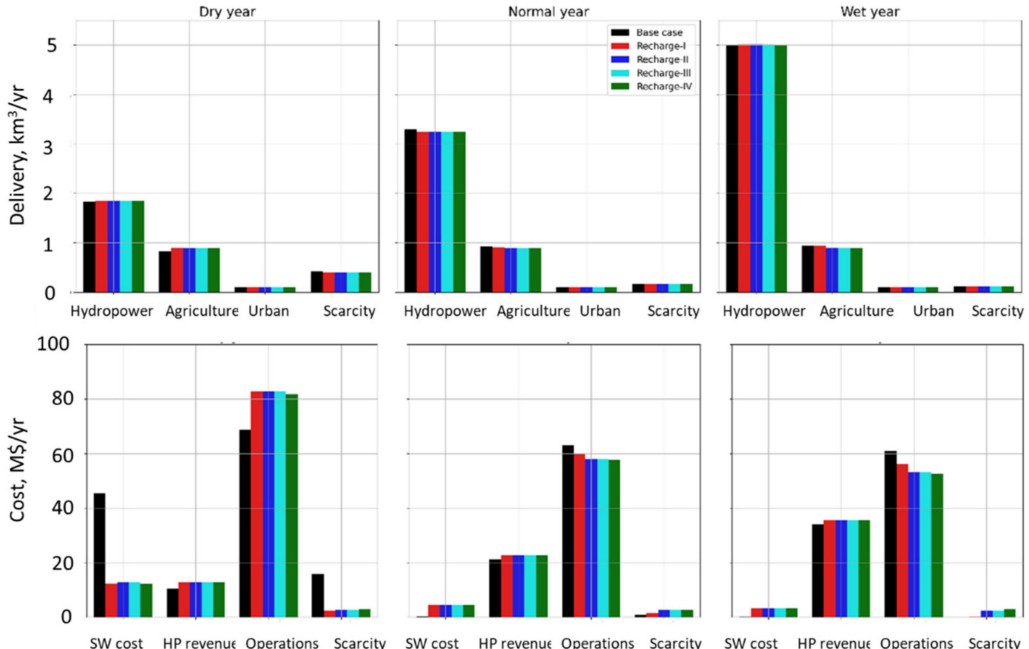

**Figure 8.** Major water supply (**top**) and cost (**bottom**) portfolio in the Lower American River in different water years.

Before developing water allocation cost portfolios, we estimated the surface water storage cost and operational cost, including hydropower revenue and scarcity cost within the region of interest. The cost of surface water storage at the Folsom Dam is obtained using Equation (8) by multiplying monthly evaporation from Folsom Lake (SR_FOL) with the average value of upper economic dual at the downstream nodes D9 and WTP204 from the lake. Here, the average upper dual is considered the opportunity cost of surface water storage based on the evaporation (see Supplemental Figure S4 and Table S7). The surface water storage cost shown in Figure S4 revealed decreased surface water storage cost in dry years and increased in normal and wet years. This is not trivial, because more evaporation from the surface water reservoir during dry years than in normal and wet years hence reduces pressure on increasing storage capacity.

Likewise, operational costs include the cost of groundwater delivery, i.e., pumping to agriculture and urban users, and wastewater and water treatment cost (Figure S3). Figure S5 shows the breakdown of operation costs, indicating increased water delivery cost (agriculture and urban) in dry and normal years. However, the agricultural delivery costs decrease more in wet years, specifically in the Recharge IV scenario, as more water is allocated for recharging than delivery. In addition, the decrease in urban water delivery

cost is smaller in normal and wet years. However, the water treatment cost is high in normal and wet years when adding a recharge facility.

The water allocation cost portfolio incorporates surface water storage and operation costs (Figure 8, bottom). Figure 8 (bottom) also shows the decreased water scarcity cost in dry years and increased scarcity cost in normal and wet years.

Economic benefits and costs for all MAR scenarios are summarized in Table 10, where surface water storage, groundwater pumping, surface water storage cost, scarcity volume, and scarcity cost seem optimistic metrics in dry years. As reported, the recharge facility can increase surface water storage as high as 0.451 km$^3$ in dry years, which is 125% more than the base case. Consequently, a recharge facility decreases the surface water storage cost by MUSD 33 in dry years even though it is increased by MUSD4 annually in normal years and MUSD 3 in wet years. The increase in hydropower generation can be as high as 85.6 MWh (MUSD 2.6 hydropower revenue) more than the base case. Scarcity Volume and cost, in dry years, are also significantly decreased by 0.067 km$^3$/yr and MUSD 12.9/yr, respectively. Despite a notable increase in operation cost by 20% in dry years, the recharge facility provides economic benefits of MUSD 34.8/yr, MUSD 1.4/yr, and MUSD 4.0/yr from the operation of water infrastructure in the Lower American River.

**Table 10.** Summary table of network system implying changes in shown metrics.

| Metrics | Dry Year | Normal Year | Wet Year |
|---|---|---|---|
| Surface water storage, km$^3$/yr | 0.451 (125%) | 0.255 (44%) | 0.149 (21%) |
| Hydropower generation. MWh/yr | 85.6 (19%) | 53.4 (6%) | 65.8 (5%) |
| Hydropower revenue, MUSD/yr | 2.6 (25%) | 1.6 (8%) | 1.5 (5%) |
| Groundwater storage, km$^3$/yr | 7.4 (3.3%) | 8.2 (3.6%) | 9.7 (4.3%) |
| Groundwater pumping, km$^3$/yr | 0.216 (23%) | −0.087 (−10%) | −0.105 (−13%) |
| Agriculture Delivery, km$^3$/yr | 0.067 (8%) | −0.011 (−1.1%) | −0.002 (−0.2%) |
| Surface water storage cost, MUSD/yr | −33.2 (−73%) | 4.1 (>100%) | 3.1 (>100%) |
| Scarcity volume, km$^3$/yr | −0.067 (−66%) | 0.027 (>100%) | 0.045 (>100%) |
| Scarcity cost, MUSD/yr | −12.9 (−82%) | 1.6 (>100%) | 2.9 (>100%) |
| Operating cost, MUSD/yr | 13.9 (20%) | −5.5 (−9%) | −8.4 (−14%) |
| Net Benefit, MUSD/yr | 34.75 | 1.43 | 3.86 |

We also summarize the optimized hydroeconomic metrics for all scenarios in dry, normal, and wet years (further details in Supplementary Tables S8–S10). Surface water storage, hydropower generation, hydropower revenue, and groundwater storage are increased in all recharge configurations across all kinds of water years. Groundwater pumping and agricultural deliveries, as well as operation costs, are increased in dry years among all recharge configurations, but such metrics are decreased in normal years and wet years. Scarcity Volume and cost, together with surface water storage cost, are significantly decreased in dry years and increased in normal and wet years. Based on hydropower revenue, scarcity cost, and operation cost, the net cost of the Lower American River region is evaluated. As seen, the system's net cost is decreased in all water years compared to the base scenario. Therefore, the recharge facility brings net annual economic benefit of MUSD 34.8, MUSD 1.4, and MUSD 3.9 in dry, normal, and wet years, respectively. The benefit of MAR is more when upstream and downstream links are partially unconstrained, i.e., no minimum recharge flow in them (Recharge IV).

## 4. Limitations and Future Research

This study does not address interrelated site-specific factors such as water availability, soil properties, cropping patterns, and the effect of groundwater quality practice. This study does not consider policy-related factors such as field-specific infrastructure, permitting or regulation. The lack of a dynamic representation in CALVIN of percolation from conveyance losses and rainfall, surface and groundwater interaction, and pumping cost variation suggests room for reflecting on the not considered additional benefits of artificial recharge.

The modeling results presented here, and subsequent analysis, focuses only on the American River Basin and encompass one surface water reservoir, one groundwater reser-

voir, and surrounding agricultural regions. Artificial recharge decreased benefits from agricultural delivery in wet years and increased operating cost in dry years; however, it decreased overall water scarcity, decreasing systemwide operation and scarcity costs. The modeling presented here can be applied to other locations and other features beyond agricultural and urban costumers can be added, such as flow to refuges, water recycling and interconnection to other water supply systems.

Another limitation of this study is the implementation of carryover storage functions only at one surface reservoir and groundwater basin near the study area, even though the simulations are conducted over the entire network. One can envision that employing similar carryover functions for all storage facilities improves the reservoir operation rules in an integrated manner [61]. Moreover, carryover storage functions are implemented only in the recharge configuration but not on the base runs. For this reason, the authors emphasize the analysis making a regional network to reflect the actual effects in the region in detail.

## 5. Conclusions

This research examines the impacts of adding a managed artificial recharge (MAR) facility near the Folsom Dam on water deliveries to cities and farms, groundwater and surface water storage, and hydropower revenues. The results suggest increases in surface and groundwater storage and in hydropower revenue when an MAR facility is available. We also find that the proposed MAR facility makes more groundwater available for pumping and delivery to agricultural users, particularly during dry years. The addition of the recharge facility does not impact the surface water delivery to the major instream flow locations where flow is more important for water transfer from the Delta to the southern part of the state.

The managed aquifer recharge considered in this study appears beneficial depending on its hydrologic, environmental, and economic consequences. Sustainable solutions for water-stressed aquifers help better manage and meet water demands, surface water substitution, and conjunctive use of groundwater and surface water. Similarly, additional recharge facilities can be examined in other parts of the large and intertwined California water supply network to facilitate the analyses needed under the Sustainable Groundwater Management Act. This study offers a first order approximation to the net benefits of MAR to buffer economic impacts of droughts in areas that benefit from increasingly scarce water supply.

**Supplementary Materials:** The following are available online at https://www.mdpi.com/article/10.3390/w14060966/s1: Figure S1. Service areas that are benefitted from the new recharge facility; Figure S2. Bar graphs showing average monthly groundwater storage and pumping for different scenarios in dry, normal, and wet water years; Figure S3. Monthly agricultural water delivery to service area shown in Figure S1; Figure S4. Bar graphs of surface water storage cost in dry, normal, and wet water years for different conditions; Figure S5. Breakdown of cost to operate water infrastructure facilities below the American River in different water years. Wtrtmnt: water treatment and WWTrtmnt: wastewater treatment; Table S1. Previous applications of CALVIN from its inception; Table S2. List of water infrastructure facilities within the area around the American River; Table S3. Marginal values and the opportunity cost of surface water storage in dollars per $m^3$; Table S4. Annual average deep percolation and additional recharge over different water years; Table S5. Mass balance at Folsom Dam (SR_FOL). All quantities are in $m^3$; Table S6. Mass balance at the groundwater basin (GW_08). All quantities are in $m^3$; S7 Table. Parameters to calculate surface water storage cost under different recharge scenarios; Table S8. Key hydroeconomic metrics for different scenarios in dry years; Table S9. Key hydroeconomic metrics for different scenarios in normal years; Table S10. Key hydroeconomic metrics for different scenarios in wet years.

**Author Contributions:** Conceptualization, M.L.M., L.L., W.A. and J.M.-A.; methodology, M.L.M. and J.M.-A.; formal analysis, M.L.M.; writing—original draft preparation, M.L.M.; writing—review and editing, J.M.-A., M.S.D., J.R.L. and A.S.F.-B.; visualization, M.L.M.; supervision, J.M.-A.; project administration, A.G.; funding acquisition, J.M.-A. and E.G. All authors have read and agreed to the published version of the manuscript.

**Funding:** NOAA under the Sectoral Application Research Program and the National Integrated Drought Information System (NIDIS) Commission PIER program (Funding Opportunity Number: NOAA-OAR-CPO-2019-2005530). National Science Foundation grant number 1639268. United States Department of Agriculture grant number 2018-67004-27405. California Strategic Growth Council grant number CCRP0013. University of California multicampus research program Labor and Automation in California Agriculture: Equity, Productivity, and Resilience (M21PR341).

**Institutional Review Board Statement:** Not applicable.

**Informed Consent Statement:** Not applicable.

**Data Availability Statement:** The CALVIN model's source code is available in a GitHub repository [73]. The CALVIN model's network is also available in a GitHub repository [74]. Data generated or analyzed during the study are available from the corresponding authors by request.

**Acknowledgments:** Sincere thanks go to several colleagues, including José Pablo Ortiz-Partida, H.B. Zeff, Keyvan Malek, and UC Merced Water Systems Management Lab members' constructive suggestions to improve this paper.

**Conflicts of Interest:** The authors declare no conflict of interest.

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
