# Peer review of "Managing Aquifer Recharge to Overcome Overdraft in the Lower American River, California, USA"

_water, doi:10.3390/w14060966_

Round 1
Reviewer 1 Report
Dear Authors,
This is an interesting study, but the methodology is not fully clear, so its scientific validity cannot be judged. Furthermore, the English of the Introduction is extremely poor. One wonders how this is possible for a manuscript with eight authors?
General comments
- Please add units for all variables in equations.
- Please add a table with the values used for all model variables. Please specify and justify your constraints for urban, agriculture and environmental water demand.
- Please add a summary table with all water balance components.
- What is the groundwater recovery efficiency for the artificial groundwater recharge facility?
- Please add a methodology section on the cost analysis (3.7).
- The water flowing into Folsom reservoir will remain the same for all scenarios. Thus, the long-term water releases should remain nearly the same, with the only differences the evaporation losses. How is it possible to have such large increases in hydropower generation?
Specific comments – Please copy-paste your modifications in the text as response to the comment.
l.21: such total costs are minimized ?
l.28: has exemplified a benchmark tool ?
l.39: worldwide and in California's Central Valley – You mean that California is not part of the world?
l.44: climate change ... overexploits ?
l.41, l.49, l.59: the recent 2012-16 drought; the recent drought; the recent 2012-2021 drought - So what recent drought does l.49 refer to? 2 billion USD in how many years? % ?
l.49: hydropower deliveries?
l.51,52 more .. more ... - more than what?
l.53: which becomes critical ?
l.55: 1200 to 25 million cubic meters (mcm) – 1200 mcm? which area, which years? add REF
l.55-57: It is not surprising that we get groundwater overdrafts due to a Sustainable Groundwater Management Act?
l.59: depleted ?
l.61: water reservoirs ?
l.61-62: by several thousands of cubic meters – 9 year drought and this is all?
l.80-81: Please rewrite
l.83: implementing ?
l.84-87: Pease rewrite
l.89: benchmark ?
l.92: Does – Can
l.95: While ?
l.99-100 that meet Delta ... ?
Figure 1: Please improve the quality of the map. Add all geographical names used in the text, including Nimbus dam and Nakatoma dam, and the main network components used in Fig 3. Explain in caption, what is meant by new? Is artificial recharge also “new”?
l.117: CVPM ?
l.123: 173 mcm of croplands ?
l.124-125: Please specify the storage capacity of the dam.
l.125: USBR ?
l.154: Please explain, is CALVIN used by the water authorities or is it a research tool or any combination of the two? Would be good to move l.201-203 here and mention if research applications have been implemented.
l.157: wildlife refuge?
Equations 1-3: Please add units for all variables. Unclear what is k, it should be an index? Why is there no time index? Please explain equation 3, Xjik = Xijk ?
l.181: Is amplitude the correct word?
l.183-184: Error 2x
l.197: HEC-PRM ?
l.198: Pyomo ?
l.199: GLPK ?
l.200: limited foresight runs ?
l.208: primarily – what else?
l.215-216: the modification includes a diversion node (DP9) before Folsom South Canal (C173) not to impact the hydropower generation from Nimbus Dam – This is unclear, both Nimbus dam and the new recharge facility are downstream from DP9.
Figure 3: Please improve legend and/or caption. Symbols for urban and agriculture are the same? Symbols for calibration, junction, diversion are the same? Where does the water of WTP204 go to? Please clarify all acronyms (CVPM, HGR... )? Please add Nimbus dam and Lake Nakatoma in the figure.
Eq. 4 and 5: There is no S in Eqs 1-3.
l.305: water types ?
l.347-348: equivalent monthly pricing seasonality ?
l.342-361: The water flowing into Folsom reservoir will remain the same for all scenarios. Thus, the long-term water release should remain nearly the same too, with the only differences the evaporation losses. How is it possible to have such large increases in hydropower generation?
l.403: swelling ?
Reviewer 2 Report
This research investigates the impacts of adding an artificial recharge facility below the Lower American River Basin on water deliveries to cities, agriculture, groundwater and surface water storage, hydropower revenues, and agricultural water use by using the hydro-economic model, CALVIN. This work demonstrates a good technical contribution and provides valuable insights. The paper is well-written and structured. My comments are mainly on minor modifications to the structure of the paper.
Major comments
Abstract:
The abstract needs to be slightly revised explaining the major research gap and its importance for a broader audience rather than directly moving into a site-specific methodology.
Introduction:
The objectives of the research can be slightly modified for a broader audience. The research questions/objectives are very site specific. Here, the novelty and the technical significance of your work should be highlighted.
Methods:
An overview to the methodology at the beginning of this section will help the readers to follow your paper, especially since your paper discussed a number of different things. A flowchart showing an overview can also help.
Please explain why you have chosen CALVIN over other models? What are the strengths and weaknesses of CALVIN?
Discussion:
I suggest authors adding a section explaining the major technical significance of your work to the scientific community (which is not site specific), how your method is important for a broader audience, limitations of your work and proposed expansions/future work into the discussion.
Abbreviations:
Each abbreviation should be expanded when it first appears in the paper. Check the whole paper carefully and fix this.
Tables:
Please add the units in Table headings instead of stating it in the Table captions.
Minor comments:
Line 35-38 - Combine the first two sentences to avoid redundancy.
Line 39 - State the country to specify the study area for an international audience.
Line 49-50 - State the period of the recent drought at this point. The term 'recent' is vague in a paper. Anyone who reads the paper in the future can get confused.
Line 55 - capitalize the unit 'mcm' throughout the paper.
Line 58 - 'drought periods'. Are you referring to California? Please specify.
Line 66-67 - Change 'Flood-MAR (Managed Aquifer Recharge)' to 'Flood-MAR (Managed Aquifer Recharge) concept'
Line 91-93 Combine this with the previous paragraph.
Line 104 Provide some climatological information of the study area, e.g., rainfall, evaporation rates, dryness/wetness, climatic zone, etc.?
Line 105-107 Please provide more details on the location w.r.t. the country for the international audience, e.g., State.
Line 106- Please check whether there's an extra space before the term 'Figure 1', and remove it if there's any.
Line 183-185 "Error! Reference source not found." - Please check
Line 202-203 Change 'Table S1' to 'Table 1'
Line 204 Including an overview to the entire methodology (optionally you can add a flow chart) will make it easier for the readers to follow the paper.
Line 223- What is the basis for assuming conveyance losses of 5%?
Line 226 - Please define Q in table caption or in a footnote (as you've defined NOB).
Line 238 – Briefly explain how you have selected the recharge months at this point of the paper.
Line 305 - WHat is the criteria of defining dry, normal and wet year?
Line 322 - Add 'month-1' to the Y-axis label of the top row of Fig. 4.
L 326-327 - I don't see an increase in the release from October to March at all recharge scenarios in Fig. 4.
L 529 - I suggest adding a section explaining the elaborating the major technical significance of your work to the scientific community (which is not site specific), limitations of your work and proposed expansions into the discussion.
Round 2
Reviewer 1 Report
Dear Authors,
The major comments below are a partial repeat from the previous review because your answers were not adequate.
(1) Ref. 57 doesn't have a doi or web address. So please specify all model constraints and parameter values (storages, demands, costs, losses, amplitudes, environmental flow requirements) in tables. Some data have been given in the Supplement, but Tables S4, S6, S7, S8 are not even referenced in the text. Also, these tables don't specify the data source! I also wonder how a paper submitted in 2021 uses data from a report in 2001? Nothing changed after all these droughts? This can't be right.
(2) Figure 8 doesn't give water balance components, it gives water supplies to different sectors. Please tabulate water balance components.
(3) I can't find the 5% losses for artificial recharge in the paper. It needs a reference and explanations why these losses are expected to be so incredibly low. Please specify the line numbers where you made the corrections and additions.
(4) The water storages and the water releases of Folsom dam, upstream of the new recharge facility, are always larger than the base scenario, in dry, average and wet years (Fig. 3, Table 3). How is this possible, considering that the inflows remains the same?!
Specific comments (something wrong with the grammar)
l.164: The model’s versatility and robustness, researchers have been shown ?
l.377: Wetter the climates have, more water is conveyed ?
l.391: The recharge facility unconstrained in both links (Recharge-I) seems ?
l.418: under the, implied by the Recharge-I scenario. ?
l.435: an additional recharge facility provides a venue for improving prospects for groundwater sustainability facilitates water transmission
l.498: the average upper dual ?
l.561: the authors emphasize the analysis making a regional network to reflect the actual effects in the region in detail ?
Round 3
Reviewer 1 Report
Dear Authors,
Thanks very much for all the additions and corrections. It is starting to become clear. I still remain a bit puzzled about the below points.
(1) The two sentences below seem to contradict each other?
l.259: This study assumes no recharge during dry years,
l.336: Tables S3 and S4 report details of water balance at the Folsom dam (SR_FOL) and groundwater basin (GW_08) for the drought year 1987-88,
(2) The below is unclear. Outflow and storage in the dam are higher for the recharge case than for the base case, but the penalty for storage is higher for the dam than for the recharge facility.
l.274: The ending groundwater storage is penalized at $0.08/m3 while ending surface water storage is penalized at $0.73/m3.
l. 339: Table S3 reveals how outflow and storage in the recharging case are higher than in the base scenario. Such implies that the imposed penalty at the upper bound of a storage facility that precludes changes in storage is mostly less in the recharge facility (Eq. 4).
(3) There seems to be something wrong with the numbers and units. Also, the comma in 62,5000 in line 257 is in the wrong place.
Table 3 gives average Q in km3/mo for Nov-March, which sums to 224.78 km3 = 224.78 x 10^9 m3 for the year.
l.257: Consequently, more water (averaging about 62,5000 m3/yr) is available for winter (November March) recharge periods.
Table 3 gives a minimum flow for Nov: 3.53 km3 = 3.53 x 10^9 m3
Table S4 gives inflow to groundwater basin in Nov 1987 (drought year): 50,860 m3
(4) Table S3: why is evaporation from the dam zero in Sep 1988?
